# Copper Death Inducer, FDX1, as a Prognostic Biomarker Reshaping Tumor Immunity in Clear Cell Renal Cell Carcinoma

**DOI:** 10.3390/cells12030349

**Published:** 2023-01-17

**Authors:** Aimin Jiang, Juelan Ye, Ye Zhou, Baohua Zhu, Juan Lu, Silun Ge, Le Qu, Jianru Xiao, Linhui Wang, Chen Cai

**Affiliations:** 1Department of Urology, Changhai Hospital Affiliated to Naval Medical University (Second Military Medical University), Shanghai 200433, China; 2Wuxi School of Medicine, Jiangnan University, Wuxi 214122, China; 3Department of Orthopedic, Changzheng Hospital Affiliated to Naval Medical University (Second Military Medical University), Shanghai 200003, China; 4Vocational Education Center, Naval Medical University (Second Military Medical University), Shanghai 200433, China; 5Department of Urology, Jinling Hospital Affiliated to Medical School of Nanjing University, Nanjing 210093, China; 6Department of Special Clinic, Changhai Hospital Affiliated to Naval Medical University (Second Military Medical University), Shanghai 200433, China

**Keywords:** copper death regulators, FDX1, tumor immunity, multiomics, renal cancer

## Abstract

Background: Progress in the diagnosis and treatment of clear cell renal cell carcinoma (ccRCC) has significantly prolonged patient survival. However, ccRCC displays an extreme heterogenous characteristic and metastatic tendency, which limit the benefit of targeted or immune therapy. Thus, identifying novel biomarkers and therapeutic targets for ccRCC is of great importance. Method: Pan cancer datasets, including the expression profile, DNA methylation, copy number variation, and single nucleic variation, were introduced to decode the aberrance of copper death regulators (CDRs). Then, FDX1 was systematically analyzed in ccRCC to evaluate its impact on clinical characteristics, prognosis, biological function, immune infiltration, and therapy response. Finally, in vivo experiments were utilized to decipher FDX1 in ccRCC malignancy and its role in tumor immunity. Result: Copper death regulators were identified at the pancancer level, especially in ccRCC. FDX1 played a protective role in ccRCC, and its expression level was significantly decreased in tumor tissues, which might be regulated via CNV events. At the molecular mechanism level, FDX1 positively regulated fatty acid metabolism and oxidative phosphorylation. In addition, FDX1 overexpression restrained ccRCC cell line malignancy and enhanced tumor immunity by increasing the secretion levels of IL2 and TNFγ. Conclusions: Our research illustrated the role of FDX1 in ccRCC patients’ clinical outcomes and its impact on tumor immunity, which could be treated as a promising target for ccRCC patients.

## 1. Introduction

According to statistics produced by the International Agency for Research on Cancer, an estimated 19.3 million new cancer cases and 10 million cancer deaths occurred worldwide in 2020 [1]. Cancer ranks as the first or second leading cause of death in most countries, placing a heavy burden on global public health [2,3,4,5]. Although tumor screening and treatment are constantly improving, almost no malignant tumors can be completely cured due to the heterogeneity and aggressiveness of cancer [6,7,8]. Clear cell renal cell carcinoma (ccRCC, or KIRC) accounts for the predominant pathological type of renal cell carcinoma (RCC), accounting for nearly 80% of all RCC cases [9,10]. Even next-generation and genomic sequencing technologies have promoted the understanding of ccRCC, and immunotherapy with combination approaches has significantly alleviated patient survival quality; only a small group of patients could receive benefits, and the heterogeneity of ccRCC remains largely unknown [11,12,13]. It is urgent to identify novel diagnostic markers and therapeutic targets to improve the outcome of patients with cancer and decode such a Pandora’s box of ccRCC.

Recently, researchers have found that excessive copper concentrations could induce cell death by targeting lipoylated TCA cycle proteins, which showed that the abundance of lipoylated proteins and FDX1 was significantly related to tumor vulnerability to treatment [14]. Ferredoxin 1 (FDX1) is located on chromosome 11q22 and encodes a low molecular mass protein containing iron-sulfur (Fe-S) clusters as a redox active group, which reduces mitochondrial cytochrome P450 enzymes [15]. The mitochondrial electron transport chain is closely related to metabolic pathways and the inflammatory response, which play an important role in carcinogenesis. Studies have identified that FDX1 participates in the synthesis of various steroid hormones and may participate in the function of granulosa cells, which is essential in the normal follicular maturation process [16]. Recently, it was reported that FDX1 is the direct target of copper-dependent cell death, which suggests a strategy to mitigate proteasome inhibitor resistance [17]. Nevertheless, much less is known about the role of FDX1 in cancer, especially in ccRCC.

Given the complexity of tumorigenesis, it is essential to investigate the role of CDRs and FDX1 at the pancancer level to evaluate their impact on patients’ clinical outcomes and the underlying molecular mechanisms. Therefore, we used multiple databases, such as TCGA and cBioPortal, to analyze CDRs and FDX1 expression aberrance, genomic alteration, biological function, and tumor immunity across different types of tumors. We also constructed FDX1-overexpressing ccRCC cell lines to verify the impact of tumor immunity and tumor malignancy. In summary, our work provides a distinctive and promising biomarker or target for cancers and ccRCC patients, which fuels the clinical surveillance, prognostic assessment, and precise treatment of ccRCC.

## 2. Method and Materials

### 2.1. Data Collection and Processing

Expression, DNA methylation, copy number variation, somatic mutation profiles and clinical pancancer information, including ccRCC, were collected from The Cancer Genome Atlas (TCGA) database. Expression profiles and clinical outcomes, including therapy naïve patients (GEO167573, GSE29609, GSE2251, GSE159115, GSE171306, ICGC-EU, E-MTAB-1980), were downloaded from the Gene Expression Omnibus (GEO), International Cancer Genome Consortium (ICGC), and Array-Express databases [18,19]. For transcriptome profiles from TCGA-KIRC, E-MTAB-1980, and ICGC-EU datasets, RNA-sequencing data (FPKM values) were transformed into transcripts per kilobase million (TPM) values, and then z-score normalization was performed. For array raw data, including GEO167573, GSE29609 and GSE2251, microarrays were processed using the RMA algorithm for background adjustment in the Affy package; z-score normalization was performed for all the gene expression profiles. Raw count profiles were utilized to perform differential expression analysis.

### 2.2. DNA Methylation and RNA Modification Analysis

To investigate the potential regulation of FDX1 expression based on genomic and epigenomic alterations, we downloaded DNA methylation and mRNA expression data from the TCGA repository. Correlation of methylation level, RNA modification regulators, and FDX1 expression was calculated by Spearman coefficient. DNA methylation sites of FDX1 in the genome included 23 different sites, and the RNA methylation pattern analyzed in this work included m1A, m5C, and m6A.

### 2.3. Enrichment Analysis

To investigate the biological role of FDX1 in ccRCC, we divided ccRCC patients into FDX1^high^ and FDX1^low^ subgroups according to the median expression level of FDX1. After calculating differentially expressed genes (DEGs) and the most correlated genes of FDX1 based on the mRNA matrix, we performed Gene Ontology (GO), Kyoto Encyclopedia of Genes and Genomes analysis (KEGG), gene set enrichment analysis (GSEA), and gene set variation analysis (GSVA) via the R packages ClusterProfiiler and GoPlot [20]. For correlation analysis of FDX1 and other signatures, we applied order rank-based analysis (ORA), which selected the overlapping FDX1-correlated signature from different ccRCC datasets, including E-MTAB-1980, GSE167573, GSE22541, GSE29609, ICGC-EU, and TCGA-KIRC, to identify the potential biological role of FDX1.

### 2.4. Immune Infiltration and ICI Response Analysis

We utilized multiple immune cell infiltration algorithms to calculate cellular components or immune cell enrichment scores in ccRCC patients to compare tumor microenvironment components between the FDX1^low^ and FDX1^high^ subgroups [21,22,23,24]. The impact of FDX1 mutation on immune cell infiltration in ccRCC was analyzed by the TIMER. The R package ESTIMATE was used to evaluate the stromal and immune scores based on ccRCC tissue expression profiling. The Tumor Immune Dysfunction and Exclusion (TIDE, http://tide.dfci.harvard.edu/ (accessed on 15 November 2022)) algorithm was used to compare immunotherapy responses between subgroups [25].

### 2.5. Drug Sensitivity Analysis

We evaluated ccRCC patients’ susceptibility to chemotherapy and molecular drugs according to the Genomics of Cancer Drug Sensitivity (GDSC, https://www.cancerrxgene.org/ (accessed on 15 November 2022)), cancer therapeutic response portal (CTRP, https://portals.broadinstitute.org/ctrp.v2.1/), and Pictorial Representation of Self and Illness Measure (PRISM, https://depmap.org/portal/download/ (accessed on 15 November 2022)) databases. The R package pRRophetic was applied to calculate the half-maximal inhibitory concentration (IC50) and cross-validate the estimated results [26]. In addition, the CellMiner (https://discover.nci.nih.gov/cellminer/home.do (accessed on 15 November 2022)) database was used to analyze the correlation of FDX1 expression and therapy sensitivity at the cell line level. Specifically, a positive correlation means that high expression of the gene indicates resistance to the drug, and low expression of the gene indicates sensitivity to the drug.

### 2.6. Copy Number Alteration Analysis

Genomic alterations in FDX1 across cancers were analyzed in the cBiPortal (https://www.cbioportal.org/ (accessed on 15 November 2022)). Somatic alterations in ccRCC between the FDX1^low^ and FDX1^high^ subgroups were analyzed, compared, and visualized via the R package Maftools [27]. Analysis of recurrent extensive and focal somatic copy number alterations (SCNA) among the genome was performed using the GISTIC 2.0 algorithm [28].

### 2.7. Verification of FDX1 Differential Expression Levels

RT-qPCR was utilized to validate FDX1 expression in paired tumor and adjacent renal tissues (including 22 ccRCC tissues from Changhai Hospital). Primer sequence information was as follows: primer for FDX1 (forwards primer: TTCAACCTGTCACCTCATCTTTG, reverse primer: TGCCAGATCGAGCATGTCATT), and primer for GAPDH (forwards primer: GGAGCGAGATCCCTCCAAAAT, reverse primer: GGCTGTTGTCATACTTCTCATGG). Antibodies for detecting different FDX1 protein levels between normal and tumor tissues were purchased from Abcam Company (ID: ab47267).

### 2.8. Overexpression of FDX1 and Coculture Experiments

The renal carcinoma cell lines (786-0 and ACHN) were purchased from the American Type Culture Collection (ATCC, Manassas, VA, USA) and cultured according to a standard protocol. FDX1 overexpression and negative control plasmids were synthesized by Shanghai GeneChem Co., Ltd. We utilized Lipofectamine iMAX to transfect cell lines with plasmids and harvested infected cells via puromycin culture for 3 weeks. CCK8 kits were adopted to test the proliferation difference between the control and FDX1 over expression treated ccRCC cell lines. We collected peripheral venous blood from healthy volunteers and stored it in a heparin anticoagulant tube. We then added Ficoll lymphocyte separation solution into the tube to dilute the blood. The PBMC were retrieved from the isolated lymphocytes. The recombinant human IFN-γ, IL-2, and CD3 monoclonal antibodies were added into PBMC suspended in serum-free medium to obtain cytokine-induced killer cells (CIK). Activated CIK were cocultured with FDX1-overexpression or control ccRCC lines for 32 h at a ratio of 6:1. The collected supernatant was collected for the enzyme-linked immunosorbent assay (ELISA) to quantify IFN-γ and IL-2 production (R&D System, Minneapolis, MN, USA) according to the manufacturer’s instructions. Absorbance was detected at 490 nm via a BioTek microplate reader. For the detailed procedure of the experiments, refer to our previous studies [29,30,31,32].

### 2.9. Statistics Analysis

All data processing, statistical analysis, and plotting were analyzed using R software (version 4.0.4). Differences between the FDX1^low^ and FDX1^high^ subgroups were analyzed by student T or Wilcoxon tests. Different expression levels of FDX1 among various in T, Stage, Grade were tested via Kruskal-Wallis test by R package ggpubr. Differences in clinical characteristics and inhibitor response were compared by the chi-square test. The influence of FDX1 on patient prognosis, including OS and PFS, was compared by Kaplan-Meier and log-rank tests. The hazard ratio (HR) was calculated by univariate Cox regression and multiple Cox regression algorithms. A two-way *p*-values test was performed, and *p* < 0.05 was considered statistically significant.

## 3. Result

### 3.1. Landscape of Copper Death Signatures across Cancers

In this research, ten copper death regulators (CDRs), including seven positive hits, ferredoxin 1 (FDX1), lipoic acid synthetase (LIAS), lipoyltransferase 1 (LIPT1), dihydrolipoamide dehydrogenase (DLD), dihydrolipoamide S-acetyltransferase (DLAT), pyruvate dehydrogenase E1 subunit alpha 1 (PDHA1), and pyruvate dehydrogenase E1 subunit beta (PDHB), and three negative hits, metal regulatory transcription factor 1 (MTF1), glutaminase (GLS), and cyclin-dependent kinase inhibitor 2A (CDKN2A), were identified. Since copper-induced cell death has gradually become a new target for cancer therapy, we first investigated the implications of these signatures at the pancancer level. As Figure 1A indicates, most CDRs were aberrantly dysregulated in cancer. Among these genes, FDX1 and CDKN2A were significantly down- and overexpressed in CHOL and CESC, respectively (Figure 1B). Regarding the survival impact of CDRs, we found that even though most CDRs were downregulated in cancers, different CDRs displayed distinctive prognostic influence (Figure 1C). In detail, nearly all CDRs exerted protective effects in KIRC and KIRP, while risk effects were observed in KICH, which indicated that the copper death signature led to different biological functions according to the cancer pathological phenotype. Furthermore, we calculated the copper death signature enrichment score based on ssGSEA algorithms and found that all cancer types led to a lower CDR score than normal tissues, and this phenomenon was most obvious in KIRC, which indicated that the dysregulated copper death pathway might facilitate the progression of tumors and that activation of the copper death signature could be treated as a new therapeutic weapon for ccRCC patients (Figure 1D). To determine the potential regulation of CDRs in cancer, we employed multiomic datasets, including DNA methylation, somatic copy number variation (CNV), and genetic variation, to investigate the aberration of CDRs. CNV displayed a high frequency in CDRs in most cancer types, among which the expression level of LIPT1 displayed a positive correlation with CNV events, and other CDRs exerted heterogeneous relationships according to cancer types (Figure 1E). CDKN2A led to the highest mutation frequency in most cancer types, and UCEC had the highest mutation frequency among all cancer types (Figure 1F). The genome location of CDRs is depicted in Figure 1G, and it should be mentioned that PDHA1 was located at chromosome X. Finally, we used the ssGSEA algorithm to analyze the potential influence of the copper death signature on cancers, which showed that this signature positively correlated with xenobiotic metabolism, oxidative phosphorylation, fatty acid metabolism, and adipogenesis, but negatively correlated with mitotic spindle, G2/M checkpoint, and epithelial mesenchymal transition (Figure 1H). All these results indicate that genomic aberrance of CDRs is common in most cancer types and further regulates the dysregulated copper death signal in cancer initiation and progression.

### 3.2. Genomic and Posttranscriptional Modification of FDX1

Regarding the detailed genomic aberration, we found that the structural variant, amplification, and deep deletion frequency of FDX1 varied among different cancer types; amplification and deletion were the main mutation patterns in ccRCC (Figure 2A). We further analyzed the type and site of the FDX1 alteration. As shown in Figure 2B, missense mutation of FDX1 was the main type of genetic alteration. In ccRCC patients’ genomic alteration landscape, interestingly, we found that the FDX1^low^ subgroup led to a higher frequency of copy number variation than the FDX1^high^ subgroup in ccRCC (Figure 2C). The detailed genomic landscape difference between subgroups is depicted in Figure 2D, which indicates that FDX1^low^ led to a high mutation frequency of low-density lipoprotein receptor-related protein 2, LRP2, and gain of chromosome.

To further analyze the correlation of aberrant FDX1 expression with methylation and RNA modification, we first calculated the correlation index of methylation site beta values and FDX1 expression, which indicated that DNA methylation played a negative regulatory role in most cancer types, including ACC, BRCA, CHOL, ESCA, HNSC, KICH, KIRP, LIHC, LUAD, OV, PCPG, SKCM, TGCT, and UCEC (Figure 3A). It should be noted that methylation had no significant impact on FDX1 expression in ccRCC. Regarding RNA modification, we found that m1A, m5C, and m6A led to a positive effect on FDX1 expression across nearly all cancer types, except TGCT, DLBC, ESCA, PCPG, CHOL, LIHC, and UCS (Figure 3B). All these results indicated that the FDX1 expression level might be regulated through different approaches in various cancer types.

### 3.3. Association between FDX1 Expression and Clinical Features

Since FDX1 was reported as the initial factor to activate copper ionophore-induced cell death and its role in ccRCC remains largely elusive, we next systematically investigated FDX1 in ccRCC. First, we detected the protein interaction of FDX1 through the ComPPI dataset, indicating that FDX1 could interact with the classic ccRCC-related target HIF1A in the cytosol, nucleus, and membrane (Figure 4A). The mRNA level of FDX1 was lower in tumor tissue in the GSE167573 and TCGA-KIRC cohorts (Figure 4B). This result was validated at the protein level, and we also found the phosphorylation level at NP_004110.1-S117 was higher in tumor tissues (Figure 4C). At the single cell level, we found that FDX1 was significantly less expressed in malignant cells compared with stromal and immune cells (Appendix A). In addition, the expression level of FDX1 was downregulated with tumor progression, including TNM, stage, and grade systems (Figure 4D). Through univariate Cox regression analysis, we found that FDX1 displayed a protective role in nearly all ccRCC datasets, except OS_GSE167573, which might be explained by different sample sizes and patient enrollment standards (Figure 4E). Kaplan-Meier survival analysis further validated that higher expression of FDX1 in ccRCC predicted better OS, DSS, and PFS (Figure 4F). Through univariate and multivariate regression analysis, we found FDX1 expression level could function as an independent factor for the OS (univariable: HR 0.77 (0.67–0.88, *p* < 0.001); multivariable: HR 0.87 (0.77–0.98, *p* = 0.025)) and PFS (univariable: HR0.76 (0.67–0.88, *p* < 0.001); multivariable: HR0.87 (0.77–0.98, *p* = 0.022)) of ccRCC (Appendix A).

### 3.4. Enrichment Analysis of FDX1 in ccRCC

In this section, we decided to analyze the biological function of FDX1 in ccRCC through differentially expressed genes (DEGs) between the FDX1^low^ and FDX1^high^ subgroups. As Figure 5A indicates, DEGs were identified. Then, GO, GSEA, and GSVA were performed to decode the implication of DEGs, which were enriched in anion transmembrane transport, negative regulation of endopeptidase and peptidase activity, monovalent inorganic cation homeostasis, and sodium ion transport in biological processes; apical part of the cell, apical plasma membrane, basolateral plasma membrane, endoplasmic reticulum lumen, and collagen-containing extracellular matrix in cellular components; serine-type endopeptidase, peptidase and hydrolase activity, and active transmembrane transporter activity in molecular functions (Figure 5B). GSVA revealed that adipogenesis, fatty acid metabolism, MTORC1 signaling, and the PI3K-AKT-MTOR pathway were activated in the FDX1 low-expression subgroup, while IL6-JAK-STAT3, Wntβ-catenin, and Hedgehog signaling were activated in the FDX1 high-expression subgroup (Figure 5C). GSEA indicated that lipid metabolism, SLC-mediated transmembrane transport, and the transport of small molecules were activated in the low FDX1-expressing ccRCC subgroups (Figure 5D). In addition, we also compared the ccRCC-related transcript factor regulon score between subgroups, which showed that the FDX1^low^ subgroup led to high activation levels of HNF1A and HNF1B, while the FDX1^high^ subgroup displayed high activation levels of FOXE1, TBX18, and TP53 (Figure 5E). KEGG analysis further validated that FDX1 was involved in valine, leucine, and isoleucine degradation, fatty acid degradation, propanoate metabolism, and oxidative phosphorylation (Figure 5F) 8 × 10^−16^.

In addition, we utilized the overrepresentation analysis (ORA) algorithm to decipher the role of FDX1 across different ccRCC cohorts. The GO results illustrated that FDX1 participated in aerobic respiration, cellular respiration, and energy derivation by oxidation of organic compounds in BP; mitochondrion and cytoplasm in CC; and oxidoreductase activity and catalytic and electron transfer activity in MF (Figure 6A). KEGG analysis revealed that FDX1 might be involved in peroxisome, Notch signaling pathway, and ECM−receptor interactions (Figure 6B). GSEA and hallmark analysis also confirmed that FDX1 mainly participated in metabolic pathways, including valine, leucine and isoleucine degradation, oxidative phosphorylation, and fatty acid metabolism (Figure 6C,D).

### 3.5. Interaction of FDX1 and Tumor Immunity

Immunotherapy combining targeted therapy has become a first-line treatment for advanced unresectable ccRCC, while only a small subgroup could benefit from combined therapy. In this section, we aimed to decipher the role of FDX1 in ccRCC tumor immunity. We found that the FDX1^low^ subtype had higher expression levels of immune regulators, which expressed higher levels of CCL5, CXCR4, CCR7, CCR4, CXCR3, ENTPD1, and TNFSF14 (Figure 7A). Similarly, TME analysis found that the FDX1^low^ subgroup contained higher infiltration levels of immune cells, such as B cells, CD4+ cells, CD8+ T cells, monocytes, macrophages, stromal cells, and immune scores (Figure 7B). We also utilized the ssGSEA algorithm to further verify this difference and found that most immune cells, MDSCs, and regulatory T cells were more highly enriched in the FDX1^low^ subgroup (Figure 7C).

Through correction analysis of immune cells and FDX1 expression, we found that FDX1 expression was positively related to dendritic cells but negatively correlated with most immune cells (Figure 8A). The most significant correlation index was the coefficient index between FDX1 and MDSCs (Figure 8B). In addition, immune dysfunction and MSI scores were significantly higher in the FDX1^low^ subgroup (Figure 8C); correspondingly, the FDX1^high^ subgroup led to a higher response rate to immune checkpoint inhibitor (ICI) therapy (Figure 8D), which was verified in a real-world ccRCC cohort (Figure 8E). FDX1 expression also exhibited good performance in predicting sensitivity in two independent ICI cohorts (Figure 8F).

### 3.6. Drug Sensitivity Analysis

Since the FDX1^low^ subgroup led to a poor prognosis, we aimed to search for potential targets sensitive to this subgroup. Interestingly, we found that the FDX1^low^ subgroup was sensitive to sunitinib and crizotinib, which were adopted in clinical practice (Figure 9A). In addition, we analyzed the correlation between FDX1 expression levels and IC50 values from the CellMiner database and found that FDX1 expression was positively correlated with the IC50 values of ifosfamide, chelerythrine, pyrazoloacridine, and KPT-9274, but negatively correlated with the IC50 values of everolimus, LY-3023414, INK-128, PQR-620, and defactinib (Figure 9B). Then, we harnessed three comprehensive drug sensitivity databases, GDSC, CTRP, and PRISM, to detect the relationship between FDX1 expression and therapy sensitivity or resistance. As Figure 10A,B illustrated, high FDX1 expression displayed a consensus result of therapy resistance to AS605240-224, AST-1306-381, AZD6482-1066, AZD8186-1918, and ABT737-1910 in the GDSC database (Figure 10A,B); AGK-2, Canertinib, Afatinib, Alisertib, and BRD9647 in the CTRP database (Figure 10C); 5-methylfurmethiodide, 10-deacetylbaccatin.1, 1-naphthyl-PP1, AC-55649, 4-methylgenistein, A-366, etc., in the PRISM database (Figure 10D).

### 3.7. Immune Impact of FDX1 Overexpression in ccRCC Cell Lines

Combined with the results based on in silico analysis, in this section, we first verified the differential expression at the RNA and protein levels, which indicated that ccRCC tumor tissues had a significantly dysregulated FDX1 expression level (Figure 11A–C). We found that ccRCC cell line malignancy could be impaired after restoring the FDX1 expression level, since the proliferation rate of FDX1 overexpression (oeFDX1) was lower than that of the control group (Figure 11D). In a previous analysis of our study, we found that the FDX1 high expression phenotype was associated with an activated immune phenotype in ccRCC, and previous reports have indicated that the activation of cell death could enhance the anti-tumor efficiency in tumors. Thus, we decided to verify this hypothesis. Interestingly, the IL2 and INF-γ secretion levels were significantly higher in medium in which T cells were cocultured with oeFDX1 cell lines (Figure 11E). All these results reminded us that restoration of FDX1 normal expression levels in ccRCC could enhance tumor immunity and hamper the malignancy of ccRCC.

## 4. Discussion

The past few decades have witnessed advances in ccRCC diagnosis and therapeutic approaches [33,34,35,36]. However, therapy resistance, immune diversity, and tumor heterogeneity hinder the alleviation of patient prognosis, which has prompted researchers to identify new therapy targets for ccRCC [37,38]. Even though the application of immune checkpoint inhibitors (ICIs) has significantly prolonged the survival of ccRCC patients with advanced tumors, a lack of tumor-infiltrating lymphocytes or an exhausted immune state depressed ICI therapy efficiency [39,40,41,42]. Apart from the benefit of immune-oncology (IO) therapy for advanced ccRCC, several alternative therapies have emerged, since a subset of those patients will lose the initial efficacy or even develop resistance [43]. Until now, several clinical trials or pre-clinical agents have been carried out, including inhibitors targeting hypoxia-inducible factor (HIF), glutaminase, adenosine, interleukin-2, interleukin-15, interleukin-27, LAG3, TIGIT, ILT2/ILT4, ILT3, TREM2, OX40 agonist, batiraxcept (AVB-S6-500), adavosertib (AZD1775), DS-6000a, tyrosine kinase, and live microbiome product, while most of those agents fail in clinical application because of toxic side effects and lower generalizability [44,45]. All these findings have reminded researchers that combination or sequential therapy might elevate the response rate and life quality of advanced ccRCC patients; thus, exploring novel therapeutic targets for ccRCC is urgently needed. Emerging evidence has suggested that triggering diverse forms of cell death, including pyroptosis, necroptosis, and ferroptosis, could reverse the nonresponsive state of cancer patients [46,47,48]. The copper death signature, or cuproptosis, has been reported as a novel type of cell death that is mediated by an ancient mechanism, turning to protein lipoylation [49,50]. Recently, several promising studies have revealed the role of copper death-related signatures in multiple cancers [51,52,53,54]. Tang et al. used machine learning and bioinformatic analysis to construct a Cu-bind protein-related score for gastric cancer patients; those with low scores had higher levels of TMB/MSI and responded well to immunotherapy [55]. Chen et al. applied cuprotosis-related lncRNAs (CRLs) to predict hepatocellular carcinoma (LIHC) and revealed that tumor mutational burden survival and prognosis were greatly different between high-risk and low-risk CRLs groups [56]. Yan et al. also applied NMF cluster analysis in hepatocellular carcinoma and identified two molecular subtypes of LIHC; in addition, they determined that knockdown of LIPT1 gene expression inhibited the proliferation and invasion of hepatoma cells [54]. The role of copper death signatures and its inducer, FDX1, in ccRCC remains largely unknown. In this study, we systematically elaborated on the expression levels, prognostic impact, and biological function of CDRs and FDX1 in diverse cancer types.

First, we performed a comparable analysis of CDRs between normal and tumor tissues to investigate their aberrance at the multiomic level, including the expression profile, DNA methylation, CNV, and SNP events. It should be mentioned that nearly all CDRs were downregulated in tumor tissues, and this phenomenon could be explained by DNA hypermethylation and CNV. Nearly all CDRs functioned as protective factors in patient prognosis. In addition, the cuproptosis enrichment score was significantly low in most tumor tissues, which indicated that inhibition of cuproptosis might promote tumor malignancy. Through enrichment analysis, we confirmed that activated cuproptosis levels could positively regulate xenobiotic metabolism, oxidative phosphorylation, fatty acid metabolism, and adipogenesis, while negatively regulating the mitotic spindle, G2/M checkpoint, and EMT pathway.

Since the dysregulated state of cuproptosis was obvious in ccRCC, FDX1 functioned as the upstream target in cuproptosis. We next aimed to analyze such a target in ccRCC. All ccRCC patients were divided into two subtypes based on FDX1 expression level: FDX1^high^ and FDX1^low^ subtypes. Between subtypes, the FDX1^low^ subtype displayed poor prognosis. We also confirmed the protective role of FDX1 in multiple ccRCC cohorts, and its expression level decreased with clinical stage and pathological grade. Because of the downregulated expression level of FDX1 in ccRCC, we found that CNV and RNA methylation regulators played significant roles in FDX1 expression. Furthermore, utilizing differential expression and co-expression analysis, we found that FDX1 is involved in normal kidney function, including anion transmembrane transport and endopeptidase activity. Dysregulated FDX1 in ccRCC could promote tumor progression through metabolic reprogramming and carcinogenic pathways, such as fatty acid metabolism, PI3K-AKT-mTOR signaling, and the Myc pathway. In general, the mutation landscape might affect the expression levels of downstream targets. In this work, the mutation frequency of LRP2 was higher in the FDX1^low^ subtype. Notably, LRP2, encoding the multifunctional endocytic receptor megalin, was reported as the major mutation signature in papillary renal cell carcinoma, which was also correlated with renal disease and facio-oculo-acoustico-renal syndrome by inducing the inability of megalin R3192Q to properly discharge ligands and ligand-induced receptor decay in lysosomes [57,58]. The detailed mechanism of LRP2 mutation and FDX1 expression level needs further experiments to be verified.

Programed cell death (PCD) and non-PCD both are involved in the turnover process of cells. PCD is orchestrated and consists of apoptosis, necroptosis, pyroptosis, ferroptosis, PANoptosis, and autophagy. Numerous research studies have found that PCD could induce tumor immunity by releasing intracellular components, including exosome, ncRNA, mtDNA, cytokines, and other small molecules, thus reshaping the tumor immune microenvironment [47,59]. Cuproptosis belongs to a subtype of programed cells, which have been proved to be involved in tumor growth and metastasis [60,61]. In addition, several studies have proved that cuproptosis might regulate the tumor microenvironment, and thus influence the efficacy of immune check point blocker therapy. Liu et al. found that lower CDRs expression in HCC patients was associated with a higher infiltration of protumor immune cells [62]. The CDRs-related non-coding signatures also impacted the immune component in cancers. A risk score system of 16 CDRs-related lncRNAs by Wang et al. revealed that lung adenocarcinoma patients with high scores displayed a greater tendency toward immune escape [63]. We applied multi classic immune deconvolution algorithms to investigate the correlation between FDX1 expression and tumor immunity. The dynamic changes in tumor immune cell infiltration, immune checkpoint molecules, and immune-related pathways are pivotal for tumor immunity and play important roles in tumorigenesis and progression [64]. In our work, we calculated the correlation between FDX1 expression and tumor immunity in ccRCC, and found that the FDX1^low^ subgroup led to an immune hot phenotype, while immune impressive signatures were also high in this subtype. We found that the FDX1 expression level was negatively correlated with MDSCs and regulatory T cells, and the dysfunction score and MSI score were high in the FDX1^low^ subgroup. The TIDE results further validated the immune depression state in the FDX1^low^ subgroup. Several studies have found that the potential reasons for the failure of immune therapy are the presence of immune depression phenotype-related cells, including MDSCs, tumor-associated macrophages, and regulatory T cells [52,65,66]. Najjar et al. proved that CXCR2^+^ PMN-MDSCs are important in reducing the activity of anti-PD1 antibodies and that anti-IL1β decreases MDSCs and delays tumor growth [67]. Recent research has shown that FDX1 was associated with changes in the inflammatory response and immune microenvironment, and the prognosis of LUAD patients [68]. Further research should be focused on the detailed mechanisms of FDX1 and the ccRCC immune microenvironment in ccRCC.

Furthermore, our study also systematically analyzed the correlation of FDX1 expression and drug sensitivity to explore potential drugs targeting FDX1 for ccRCC. Interestingly, we found that the FDX1^low^ subtype was sensitive to sunitinib and crizotinib, and the FDX1 expression level was also positively affected by ifosfamide, chelerythrine, pyrazoloacridine, and KPT−9274, but was sensitive to everolimus and LY-3023414. Everolimus, an inhibitor of mTOR, has been indicated for the treatment of advanced KIRC patients who have failed treatment with sunitinib or sorafenib [69]. All these findings might provide new insight for targeting the FDX1^low^ subtype and function as inducers of cuproptosis. Moreover, in vitro experiments were carried out to verify the impressive effect of FDX1 on the proliferation, invasion, and migration of ccRCC cell lines. Combined with these results, our study indicated that FDX1 might be a target of chemotherapeutic drugs or molecular mechanisms of tumor resistance. The expression level of FDX1 was lower in ccRCC tumor tissues, and the total copper death enrichment scores were also decreased in tumor tissues. Tsvetkov and colleagues revealed that the copper death inducer, elesclomol, could restore the normal cuproptosis state in lung carcinoma cell lines through lipid peroxidation [17]. We thus hypothesized that the use of elesclomol might be the proper approach for ccRCC therapy based on cuproptosis restoration, instead of the over-expression of FDX1, which might cause off-target effects of in situ overexpression.

Although this work revealed some findings between cuproptosis and pancancer, some limitations of our work should be acknowledged. First, all the samples enrolled in our study were retrospective, and fresh samples are urgently needed to perform further validation of our findings. Second, since the datasets collected to analyze the role of FDX1 in ccRCC mainly consisted of pretreatment cohorts, future validation should consider datasets based on clinical trial studies, including chemotherapy and immunotherapy combination therapy. Third, even though we found that FDX1 overexpression could lead to an immune activation phenotype in ccRCC, the detailed mechanism should be verified in more systematic experiments.

## 5. Conclusions

In summary, the results of the present study indicated that activation of cuproptosis can be treated as a novel weapon for cancer therapy, since this signal was depressed in multiple cancer types. More interestingly, FDX1 functioned as a promising biomarker and therapeutic target for ccRCC patients by reshaping tumor immunity. These findings may help to elucidate the role of FDX1 in tumorigenesis and development, which provides a reference for the realization of more precise and personalized immunotherapy for ccRCC patients.

## Figures and Tables

**Figure 1 cells-12-00349-f001:**
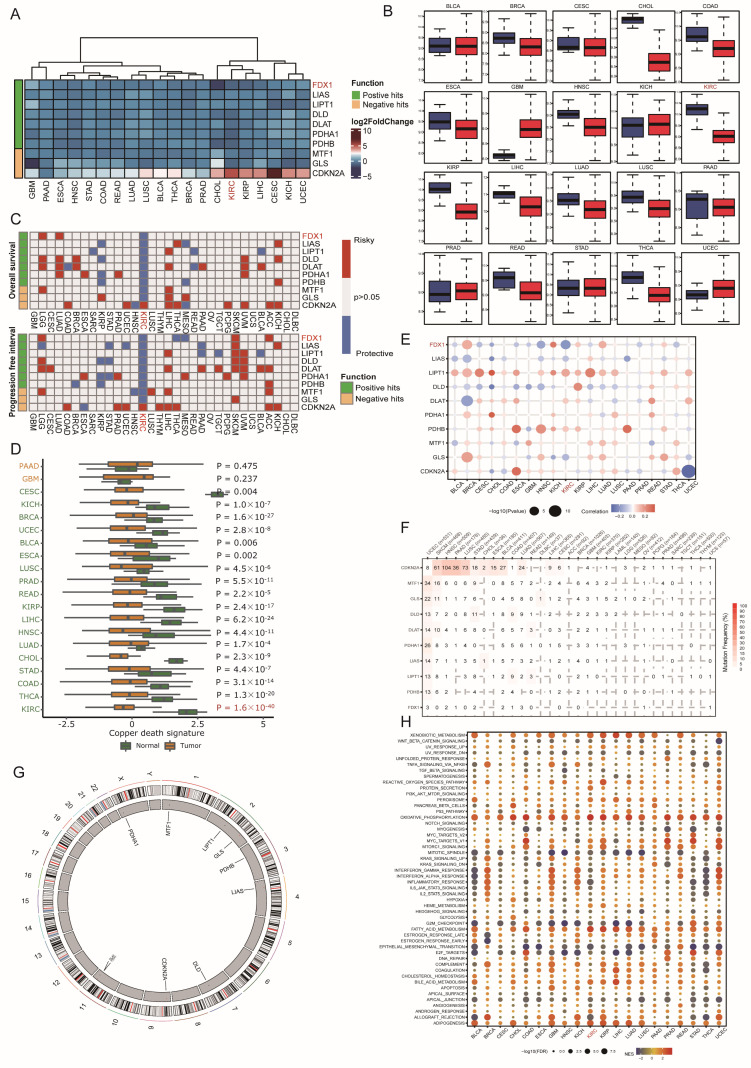
Dysregulation and mutation profile of copper death regulators in multiple cancers. (**A**,**B**) The relative gene expression of CDRs in multiple cancer tissues compared to normal tissues. (**C**) The relationship between CDRs and patient survival. (**D**) Cuproptosis enrichment score across cancers. (**E**) Correlation analysis of CNV with the gene expression of CDRs. (**F**) Mutation frequency of CDRs across cancers. (**G**) The genome locations of CDmn Rs on chromosomes. (**H**) The correlation of cuproptosis and hallmarks across cancers.

**Figure 2 cells-12-00349-f002:**
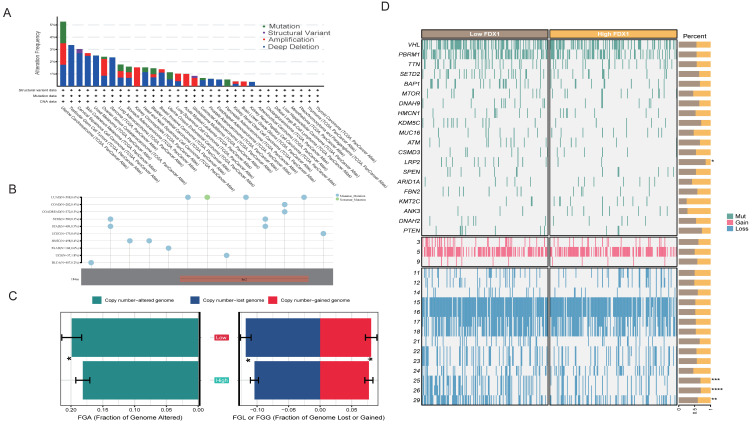
The genomic alteration of FDX1 and mutation landscape between subtypes. (**A**) FDX1 mutation pattern and frequency across cancers. (**B**) Mutation diagram of FDX1 across multiple cancers of protein domains. (**C**) Bar plot of fraction genome alterations between FDX1^high^ and FDX1^low^ subtypes. (**D**) Mutation landscape of ccRCC between FDX1^high^ and FDX1^low^ subtypes. * *p* < 0.05, ** *p* < 0.01, *** *p* < 0.001, **** *p* < 0.0001.

**Figure 3 cells-12-00349-f003:**
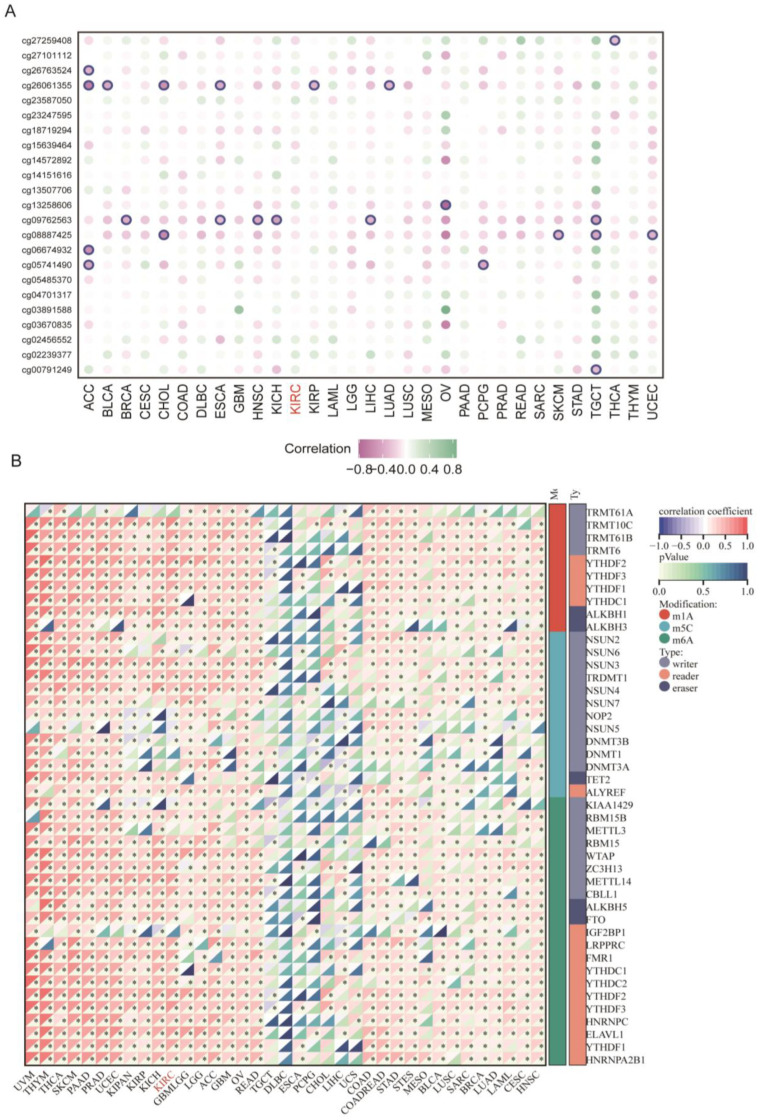
DNA methylation and RNA modification of FDX1 in ccRCC. (**A**) Correlation of different methylation sites and FDX1 expression. (**B**) Correlation of FDX1 expression and RNA methylation regulators across cancers. * *p* < 0.05.

**Figure 4 cells-12-00349-f004:**
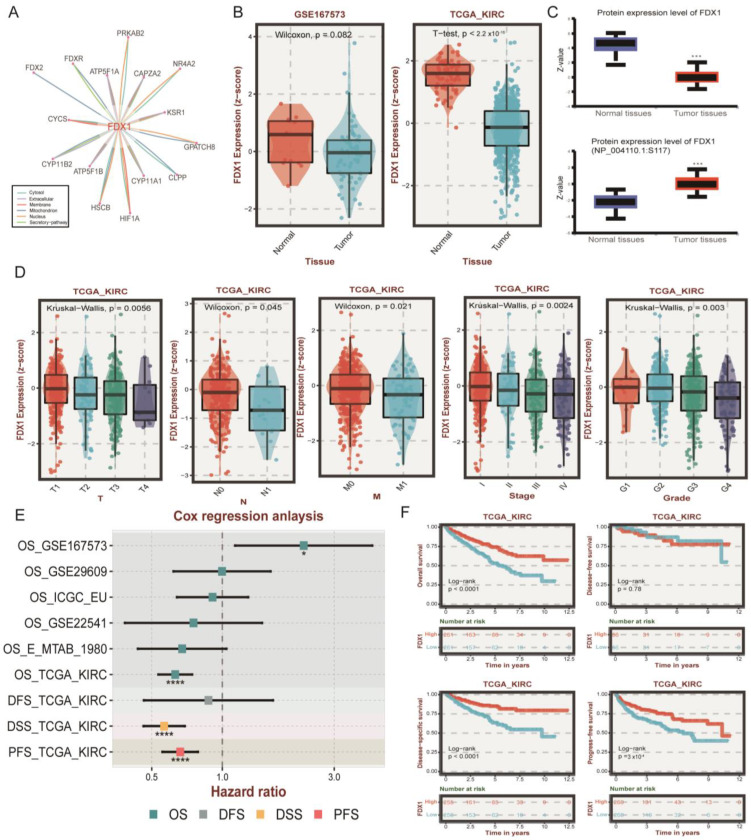
The role of FDX1 in ccRCC patient clinical outcomes. (**A**) The protein-protein interaction (PPI) network between FDX1 and other signatures. (**B**) Differential expression levels of FDX1 in GSE167573 and TCGA-KIRC. (**C**) Different protein and phosphorylation levels of FDX1 in the CPTAC database. (**D**) Relationship of FDX1 expression and clinical traits in TCGA-KIRC. (**E**) Cox analysis of FDX1 in different ccRCC datasets. ns *p* > 0.05, * *p* < 0.05, and *** *p* < 0.001, **** *p* is nearly zero. (**F**) Kaplan-Meier curves of FDX1 expression level-based subtypes in OS, DFS, DSS, and PFS.

**Figure 5 cells-12-00349-f005:**
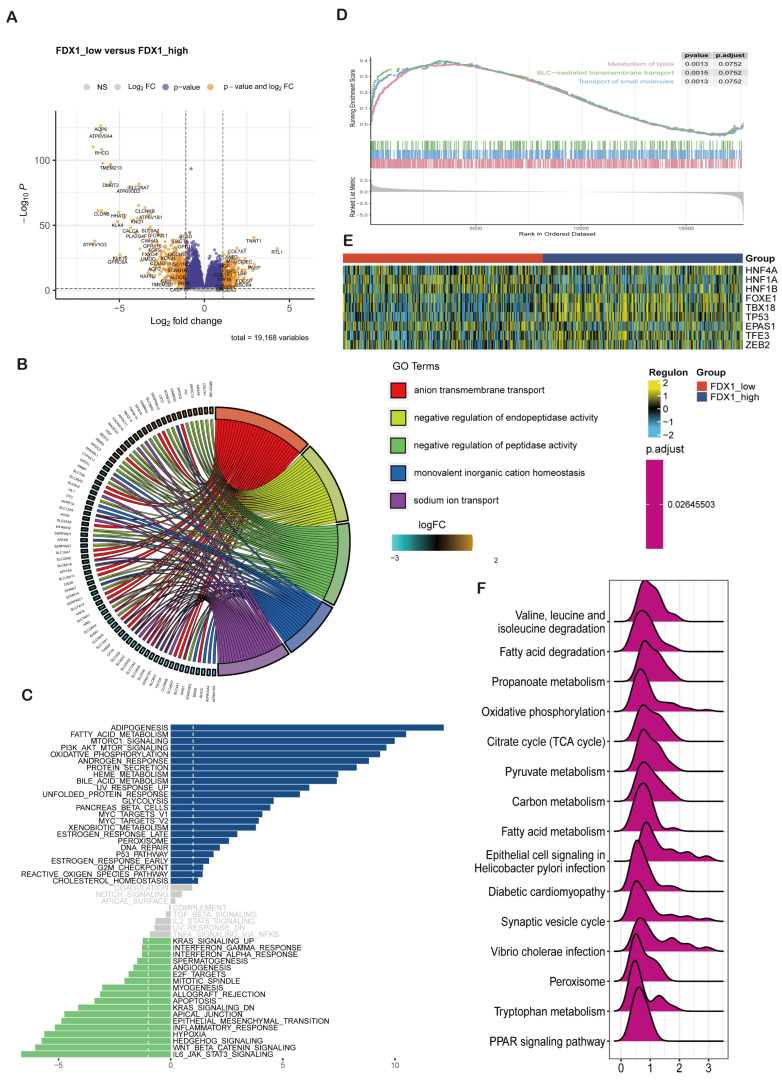
FDX1-impacted signature in ccRCC based on differential expression analysis. (**A**) Volcano plot showing differentially expressed genes between FDX1^low^ and FDX1^high^ subtypes. (**B**) GO enrichment analysis, (**C**) GSVA, (**D**) GSEA between subtypes. (**E**) Heatmap of transcription factor activation status between subtypes. Yellow represents activated expression of transcription factors. Blue represents repressed expression of transcription factors. (**F**) KEGG analysis between subtypes.

**Figure 6 cells-12-00349-f006:**
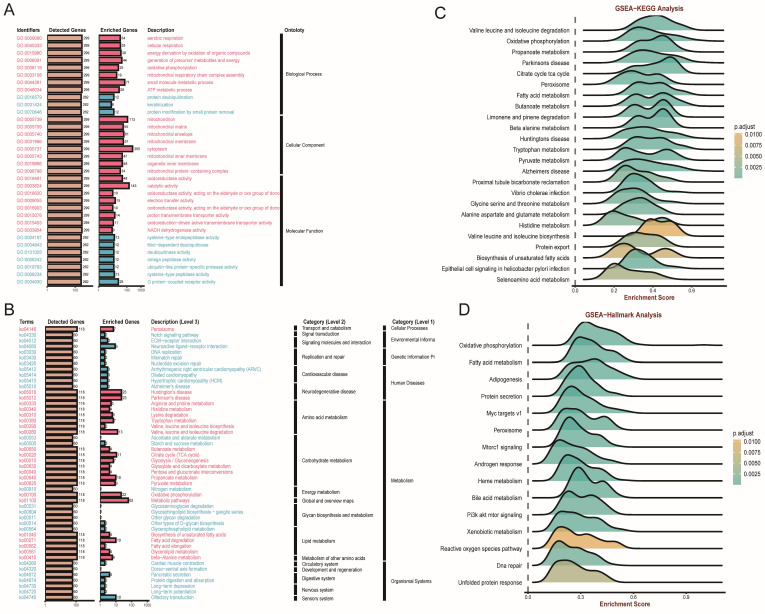
FDX1-impacted signature in ccRCC based on correlation analysis. (**A**,**B**) GO and KEGG analyses based on ORA. (**C**,**D**) GSEA-KEGG, and GSEA-Hallmarks analysis of FDX1 in ccRCC.

**Figure 7 cells-12-00349-f007:**
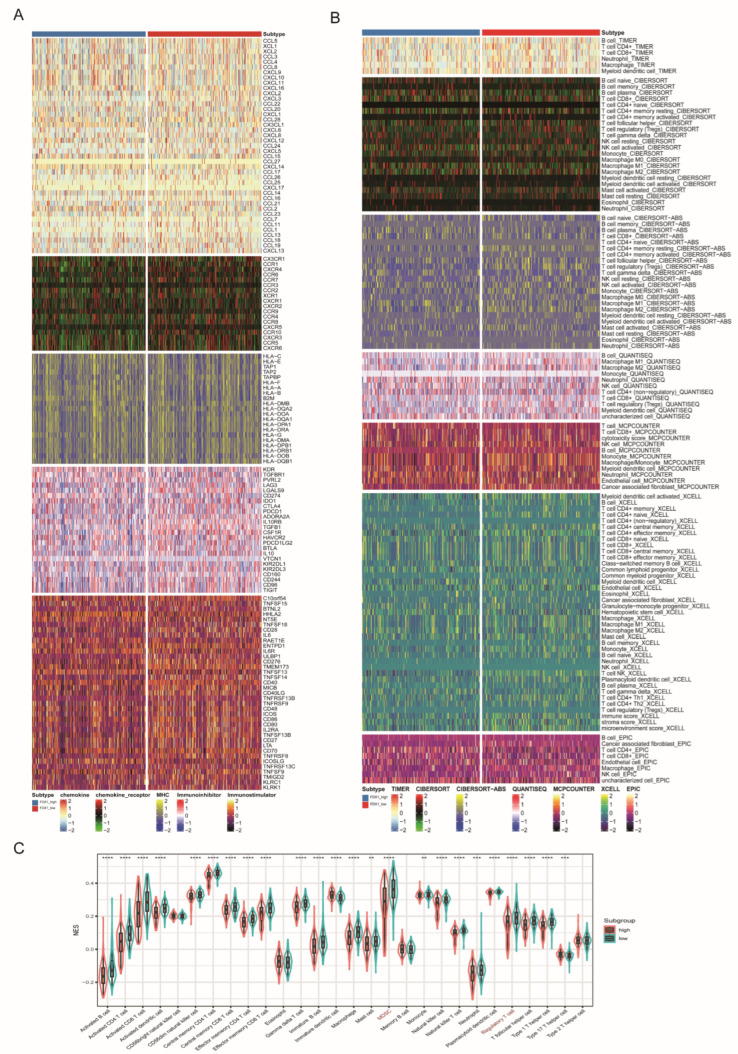
Immune profiling between FDX1^high^ and FDX1^low^ subtypes. (**A**) Heatmap of immune-related genes between subtypes, including chemokine, chemokine receptor, MHC, immunoinhibitory, and immunostimulatory. (**B**) Heatmap of tumor-related infiltrating immune cells according to immune cell quantity algorithms. (**C**) Boxplot of immune cell infiltration based on the ssGSEA algorithm. ** *p* < 0.01, *** *p* < 0.001, **** *p* < 0.0001.

**Figure 8 cells-12-00349-f008:**
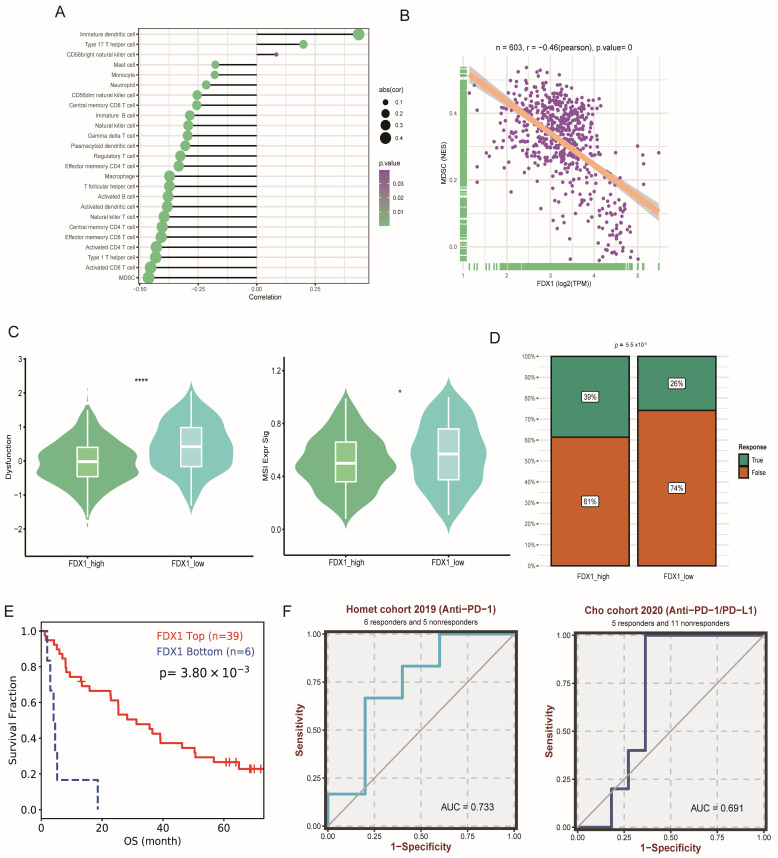
Impact of FDX1 on ccRCC tumor immunity. (**A**,**B**) Associations between FDX1 expression and the degree of immune cell infiltration in TCGA-KIRC. (**C**) The immune dysfunction and MSI score between subtypes. (**D**) Immune therapy response rate between subtypes. (**E**) Kaplan-Meier curves of FDX1 expression level-based subtypes in ccRCC ICI cohorts. (**F**) Prediction performance of FDX1 in ccRCC patients’ ICI therapy. * *p* < 0.05, **** *p* < 0.0001.

**Figure 9 cells-12-00349-f009:**
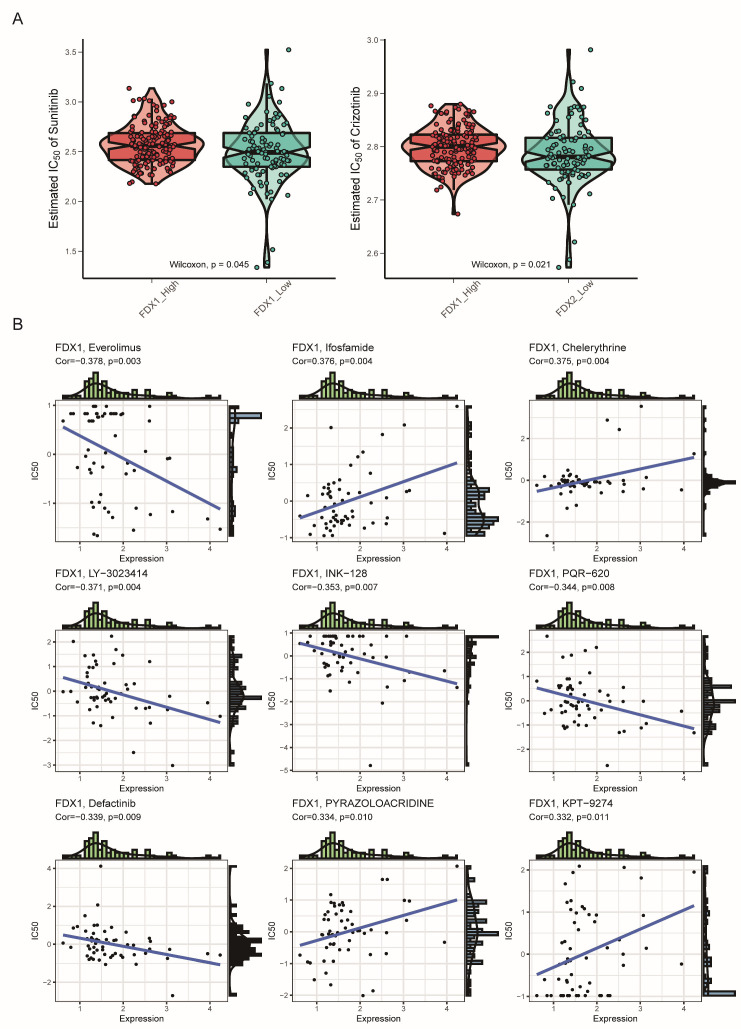
Drug sensitivity comparison between subtypes. (**A**) Estimated IC50 of the indicated molecular targeted drugs between subtypes. (**B**) Correlation of FDX1 expression and IC50 of preclinical drugs in the CellMiner database.

**Figure 10 cells-12-00349-f010:**
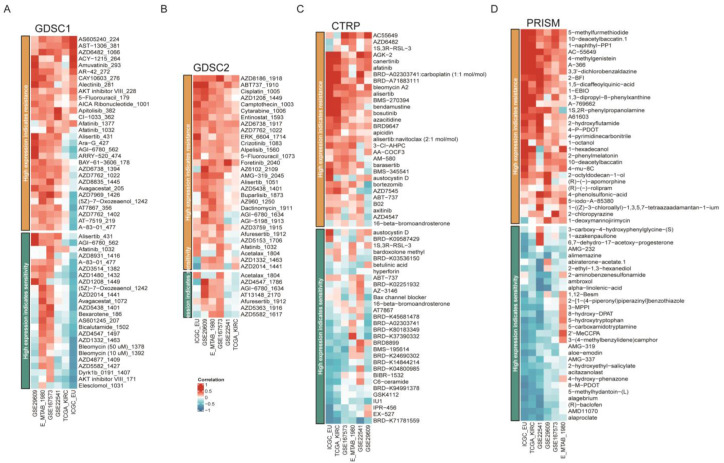
Drug sensitivity difference among different databases. Correlation of IC50 of molecular drugs and FDX1 expression levels across different databases in the GDSC database (**A**,**B**), CTRP database (**C**), and PRISM database (**D**).

**Figure 11 cells-12-00349-f011:**
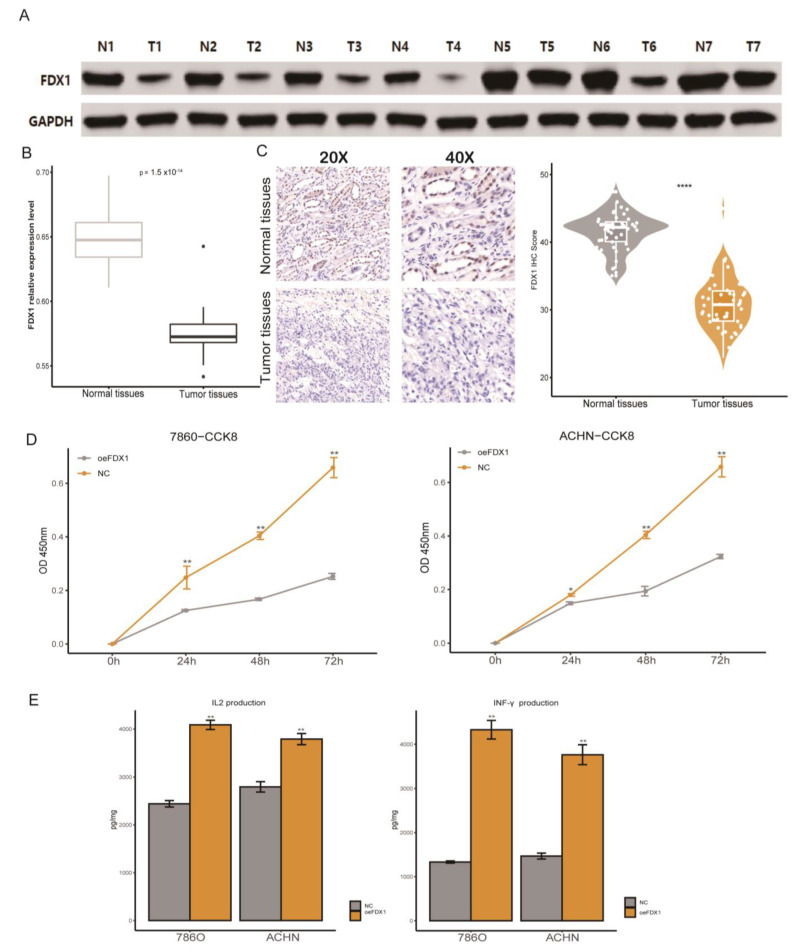
Verification of FDX1′s aberrant expression and immune impact on ccRCC. (**A**) Western blotting validated the FDX1 expression level in ccRCC. (**B**) RT-qPCR results of FDX1 expression levels in normal and tumor tissues. (**C**) Representative immunohistochemical images of FDX1 from the Changhai ccRCC cohorts. (**D**) Cell proliferation of 786-O and ACHN cells in the NC and FDX1-overexpressing cell lines. (**E**) The IL-2 and IFN-γ protein levels in the coculture medium were measured by ELISA after coincubation. * *p* < 0.05, ** *p* < 0.01, **** *p* < 0.0001.

## Data Availability

The datasets presented in the study are included in the Methods and Materials section. Further inquiries can be directed to the corresponding authors.

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
