# Peer review of "Copper Death Inducer, FDX1, as a Prognostic Biomarker Reshaping Tumor Immunity in Clear Cell Renal Cell Carcinoma"

_cells, 2023, doi:10.3390/cells12030349_

Round 1
Reviewer 1 Report
In this manuscript, the authors have explored the copper death signatures in pan cancers, especially in ccRCC. The authors focused on FDX1 and evaluated its signature in ccRCC. However, there are some remaining questions to be answered:
1, The title indicated the topic is about the role of FDX1 in ccRCC. However, the authors performed pan cancer analysis in the first parts of the manuscript. The conclusion does not seem to be consistent across all types of cancers. The results were seem consistly in ccRCC only. Since the topic is focused on ccRCC, could the authors explain why they don’t directly focused on ccRCC (KIRC) or RCC and subtypes of RCC? But instead, try to analyze the pattern in pan cancer settings?
2, There are lots of acronyms in the manuscripts. The authors should note some of them in figure legends or briefly described them. For example, what is LRP2 (mentioned 3.2)? The authors just indicated FDX1low led to a high mutation frequency of LRP2 without any explanation of LRP2 in background. In addition, they should use the acronyms consistent. For example, KIRC is equivalent to ccRCC. However, KIRC was used in figures while ccRCC was used in the paragraphs.
3, Some related references are needed for some conclusions.
4, Why the authors selected these ten CDRs?
5, When the authors divided the groups into FDX1low and high, could the authors explain how they separated the group and what the criterias were? Based on average expression, median expression? Have the authors optimized the threshold to get better prognosis in the following analysis (For example Fig 8F)?
6, The resolution of most figures are terrible and unreadable. Especially Fig 2.
7, Figure 1 missing panels notes. There is no A, B, C, D, E, and F above each panel.
8, In 3.3, the authors suggested that FDX1 could interact with the ccRCC downstream target HIF1A in the cytosol, nucleus, and membrane (Figure 4A). Could the authors explain how the interaction were performed at multiple locations in cells? Where does FDX1 express? Will FDX1 translocated to cytosol, nucleus, or secreted out of cells? How the authors determine these genes are downstream targets, not upstream regulators?
9, What does phosphorylation of FDX1 mean? How the phosphorylation of FDX1 related to protein level (Fig 4C)?
10, in Fig 4D, there was one p value in T, Stage, and Grade. Could the authors explain how they performed the statistical analysis?
11, For drug sensitivity, the authors listed many drugs in the manuscripts, are any of them have been used in clinical or pre-clinical settings for ccRCC?
12, Since FDX1low decreased the efficiency of ICI treatment and benefit ccRCC cells, have the authors knock down/out FDX1 or inhibit FDX1 via potential small molecular inhibitors and evaluate the effect?
13, The authors indicated that FDX1 could serve as a target for treatment. Is there any approaches to restore FDX1 expression in pre-clinical settings? Any potential drugs could elevate its expression in ccRCC?
Author Response
Dear Editor and Reviewers,
Thank you for your work and for our work titled as Copper Death Inducer, FDX1, as a Prognostic Biomarker Reshaping Tumor Immunity in Clear Cell Renal Cell Carcinoma (ID: cells-2109105). We feel sorry for the ignorance in our manuscript. According to your kind suggestions, we have made a point-to-point revision as follow:
Reviewer reports:
Reviewer #1: In this manuscript, the authors have explored the copper death signatures in pan cancers, especially in ccRCC. The authors focused on FDX1 and evaluated its signature in ccRCC. However, there are some remaining questions to be answered:
1, The title indicated the topic is about the role of FDX1 in ccRCC. However, the authors performed pan cancer analysis in the first parts of the manuscript. The conclusion does not seem to be consistent across all types of cancers. The results were seem consistly in ccRCC only. Since the topic is focused on ccRCC, could the authors explain why they don’t directly focused on ccRCC (KIRC) or RCC and subtypes of RCC? But instead, try to analyze the pattern in pan cancer settings?
Response: Thank you for careful consideration. Our team focused on the clinical and experimental research of genitourinary tumors, especially in renal cell carcinoma. Clear cell renal cell carcinoma (ccRCC, or KIRC) is the most common subtype of renal cell carcinoma (RCC), accounting for approximately 75% of RCC cases and the majority of deaths from kidney cancer [1]. For the availability of tumor specimen and patient’s cohort, we thus chose ccRCC, or KIRC as the main cancer type for comprehensive study of FDX1. Pan-cancer research only accounts for a very small part of the main content of this study (only in Figure1). The main analysis purpose of pan-cancer analysis is to let readers or researchers better understand the expression level, the prognostic impact, and the activation state of copper dearth regulators (CDRs) molecules at the pan-cancer level, which might provide a paradigmatic conclusion for scholars studying other cancer types. We found that nearly all CDRs (including FDX1, LIAS, LIPT1, DLD, DLAT, PDHA1, PDHB, MTF1, GLS) were lower expressed in tumor tissues compared with normal tissues, and copper death enrichment score was also lower in tumor tissues. Since the extreme heterogeneity and complexity of tumors, the prognostic impact and detailed biological role of these CDRs might vary across cancers, which need other researchers to conduct specific research. We have revised and re-written this part in Line 179-180.
Reference
- Hsieh, J.J., Purdue, M.P., Signoretti, S., Swanton, C., Albiges, L., Schmidinger, M., Heng, D.Y., Larkin, J., and Ficarra, V. (2017). Renal cell carcinoma. Nat. Rev. Dis. Primers 3, 17009. https://doi.org/10.1038/ nrdp.2017.9.
2, There are lots of acronyms in the manuscripts. The authors should note some of them in figure legends or briefly described them. For example, what is LRP2 (mentioned 3.2)? The authors just indicated FDX1low led to a high mutation frequency of LRP2 without any explanation of LRP2 in background. In addition, they should use the acronyms consistent. For example, KIRC is equivalent to ccRCC. However, KIRC was used in figures while ccRCC was used in the paragraphs.
Response: Thank you for such kind suggestions. We have added the abbreviation list of the acronyms presented in our manuscript for better understanding. LRP2 represents Low Density Lipoprotein Receptor-Related Protein 2. Both ccRCC and KIRC are the abbreviation of clear cell renal cell carcinoma. For the reason we chose ccRCC in paragraph while KIRC in figures is as follow: ccRCC represents the formal expression of clear cell renal cell carcinoma; KIRC is consistent with other cancers’ abbreviations from TCGA project in pan-cancer analysis part (such as KIRP: kidney renal papillary cell carcinoma, KICH: kidney chromophobe carcinoma). The added part could be seen in Line 227-228,527-559 (marked in Red). We hope our revision could help all readers’ better understanding of our work.
3, Some related references are needed for some conclusions.
Response: Thank you for kind reminding. We have added the related references into conclusion part. (Line 401-403, 416-417,465-480)
4, Why the authors selected these ten CDRs?
Response: The ten copper death regulators (CDRs) were adopted from studies of Peter Tsvetkov et al., who listed ten copper induced cell death related signatures (shown in the Figure bellow, in red box) with the use of Crisp Screen and numerous in vivo and vitro experiment [1].
Reference
- Tsvetkov, P.; Coy, S.; Petrova, B.; Dreishpoon, M.; Verma, A.; Abdusamad, M.; Rossen, J.; Joesch-Cohen, L.; Humeidi, R.; Spangler, R.D., et al. Copper induces cell death by targeting lipoylated TCA cycle proteins. Science (New York, N.Y.) 2022, 375, 1254-1261.
5, When the authors divided the groups into FDX1low and high, could the authors explain how they separated the group and what the criterias were? Based on average expression, median expression? Have the authors optimized the threshold to get better prognosis in the following analysis (For example Fig 8F)?
Response: We adopted the median expression level of FDX1, thus divided ccRCC patients into FDX1high and FDX1low subgroups. This part has been revised in Line 96-97(marked in Red). Such threshold could significantly distinguish different prognosis of ccRCC patients, which revealed the important clinical impact of FDX1 in ccRCC.
6, The resolution of most figures are terrible and unreadable. Especially Fig 2.
Response: We have re-uploaded all figures with high resolution in our revised manuscript (Figure 1-11).
7, Figure 1 missing panels notes. There is no A, B, C, D, E, and F above each panel.
Response: We have re-uploaded new Figure1 with panel notes in our revised manuscript.
8, In 3.3, the authors suggested that FDX1 could interact with the ccRCC downstream target HIF1A in the cytosol, nucleus, and membrane (Figure 4A). Could the authors explain how the interaction were performed at multiple locations in cells? Where does FDX1 express? Will FDX1 translocated to cytosol, nucleus, or secreted out of cells? How the authors determine these genes are downstream targets, not upstream regulators?
Response: We found the main subcellular location of FDX1 were focused on mitochondrion, then cytosol, endoplasmic reticulum, nucleus, extracellular, plasma membrane according to the Genecard database, seen in the picture below (https://www.genecards.org/cgi-bin/carddisp.pl?gene=FDX1&keywords=FDX1#localization). All those finding reminded us of the FDX1 might been translocated into different region to perform different biological roles. For the interaction relationship between FDX1 and HIF1A were retrieved from ComPPI database(https://comppi.linkgroup.hu/), the compartmentalized protein-protein interaction database, which provides qualitative information on the interactions, proteins and their localizations integrated from multiple databases for protein-protein interaction network analysis [1]. We admitted that all those findings were based database mining, which need further experiment validation. For whether these genes are downstream targets, not upstream regulators. We agreed your concern and we think our previous statement was not accurate. Thus, we revised the ‘ccRCC downstream target HIF1A’ as the ‘classic ccRCC related target HIF1A’ in Line 251. We also proved the positive correlation of FDX1 and HIF1A in Timer (left) and Gepia2 (right) database (shown in figures below).
Reference
- Veres DV, Gyurkó DM, Thaler B, Szalay KZ, Fazekas D, Korcsmáros T, Csermely P. ComPPI: a cellular compartment-specific database for protein-protein interaction network analysis. Nucleic Acids Res. 2015 Jan;43(Database issue):D485-93. doi: 10.1093/nar/gku1007.
9, What does phosphorylation of FDX1 mean? How the phosphorylation of FDX1 related to protein level (Fig 4C)?
Response: We feel sorry for the inaccurate statement between FDX1 protein level and phosphorylation level, which might largely affect the protein biological activity while not expression level. We re-write this part as: This result was validated at the protein level, and we also found the phosphorylation level at NP_004110.1-S117 was higher in tumor tissues (Figure 4C). in Line253-255.
10, in Fig 4D, there was one p value in T, Stage, and Grade. Could the authors explain how they performed the statistical analysis?
Response: For the statistical test of FDX1 expression level in different T, Stage, and Grade. Since the data distribution characteristics (There are fewer patients with advanced T, Stage and Grade), we thus applied Kruskal-Wallis to perform statistic test. And we added this part in the method and materials (Line 159-160).
11, For drug sensitivity, the authors listed many drugs in the manuscripts, are any of them have been used in clinical or pre-clinical settings for ccRCC?
Response: Thank you for you reminding. As shown in Figure9A, sunitinib and crizotinib are widely adopted in clinic for the treatment of advanced solid tumor, including clear cell renal cell carcinoma. For the results in Figure9B and Figure10, all those drugs are the pre-clinical agents, which significantly showed correlation of IC50 and FDX1 expression level, which might be potential drugs for ccRCC subgroup with low FDX1 expression level. We revised this part in Line 345-346 accordingly.
12, Since FDX1low decreased the efficiency of ICI treatment and benefit ccRCC cells, have the authors knock down/out FDX1 or inhibit FDX1 via potential small molecular inhibitors and evaluate the effect?
Response: Since the FDX1 expression level were lower in ccRCC cell lines and tissues compared with the normal counterpart. Thus, it might be optimal to restore the normal copper death state in tumor tissue with the application of copper death inducer including Elesclomol and Cucl2, which were reported in Tsvetkov’s study [1]. We have carried such experiments and aimed to investigate the correlation between copper death signatures and anti-tumor immunity in ccRCC, all those results will be presented as independent research findings. And we will share those results as a independent research.
Reference
1 Tsvetkov, P.; Coy, S.; Petrova, B.; Dreishpoon, M.; Verma, A.; Abdusamad, M.; Rossen, J.; Joesch-Cohen, L.; Humeidi, R.; Spangler, R.D., et al. Copper induces cell death by targeting lipoylated TCA cycle proteins. Science (New York, N.Y.) 2022, 375, 1254-1261.
13, The authors indicated that FDX1 could serve as a target for treatment. Is there any approaches to restore FDX1 expression in pre-clinical settings? Any potential drugs could elevate its expression in ccRCC?
Response: Thank you for such constructional and inspiring suggestions. Even FDX1 was dysregulated in ccRCC, it might be difficult to restore its expression in ccRCC tumor tissues since the limited advance of in situ overexpression technology. However, our findings revealed that the copper death enrichment score were broadly suppressed in tumor tissue across multi cancer types (shown in Figure 1D). We hypothesized that the copper death inducer agent, Elesclomol and Cucl2, which were reported in Tsvetkov’s study [1], could restore or elevate copper death state in ccRCC to reach the therapeutic purpose.
Reference
- Tsvetkov, P.; Coy, S.; Petrova, B.; Dreishpoon, M.; Verma, A.; Abdusamad, M.; Rossen, J.; Joesch-Cohen, L.; Humeidi, R.; Spangler, R.D., et al. Copper induces cell death by targeting lipoylated TCA cycle proteins. Science (New York, N.Y.) 2022, 375, 1254-1261.
We express our great appreciation again to editor and reviewers for those insightful and kind comments on our paper. Thank you for your attention and time. Looking forward to hearing from you.
Yours sincerely,
Linhui Wang
12.2022
Second Military Medical University
Shanghai, China
E-mail: wanglinhui@smmu.edu.cn; wanglinhuicz@163.com

Reviewer 2 Report
Jiang et al. address an important point, the need for novel biomarkers and therapeutic targets for ccRCC. The authors state that an aberrance in the expression of copper death regulators such as FDX1 may be important for ccRCC. They perform metanalysis on multiple data sets to identify the presence of this aberrance and its indications. However, the manuscript needs a major revision. The authors need to reformat the manuscripts regarding the display of their findings in the figures and provide a better focus for the paper. The figures presented are illegible, and the figure legends don’t clearly describe the images shown.
Major critics:
The method section needs to be written, and the authors should define what cut-offs determine the high and low expression of FDX1.
The authors show multiple datasets of different cancers and use many abbreviations; however, the information needs to be clearly presented.
The paper has multiple images that need to be clarified. The images need clear figure legends. Higher-resolution images are also necessary to understand the presented data, and some figures can be removed or combined for clearer data presentation. A description of the abbreviation in a supplementary table is also needed to help readers understand what the authors are presenting.
Specifics
The authors make multiple assumptions throughout the manuscript.
Parts of Figures 1, 2, 6, 7, 8, and 10 are illegible, and Figure 1 is missing the notations.
Figure 11 needs to be clarified regarding the methodology and the findings.
Overall, Jiang et al. did an extensive metanalysis; however, the presentation of the finds is poor. It becomes difficult to decipher the author's direction and understanding of the findings.
Author Response
Dear Editor and Reviewers,
Thank you for your work and for our work titled as Copper Death Inducer, FDX1, as a Prognostic Biomarker Reshaping Tumor Immunity in Clear Cell Renal Cell Carcinoma (ID: cells-2109105). We feel sorry for the ignorance in our manuscript. According to your kind suggestions, we have made a point-to-point revision as follow:
Reviewer reports:
Reviwer#2
Jiang et al. address an important point, the need for novel biomarkers and therapeutic targets for ccRCC. The authors state that an aberrance in the expression of copper death regulators such as FDX1 may be important for ccRCC. They perform metanalysis on multiple data sets to identify the presence of this aberrance and its indications. However, the manuscript needs a major revision. The authors need to reformat the manuscripts regarding the display of their findings in the figures and provide a better focus for the paper. The figures presented are illegible, and the figure legends don’t clearly describe the images shown.
Major critics:
The method section needs to be written, and the authors should define what cut-offs determine the high and low expression of FDX1.
Response: Thank you for kind suggestions. We adopted the median expression level of FDX1, thus divided ccRCC patients into FDX1high and FDX1low subgroups. This part has been revised in Line 96-97(marked in Red).
The authors show multiple datasets of different cancers and use many abbreviations; however, the information needs to be clearly presented.
Response: Thank you for such kind suggestions. We have added the abbreviation list of the abbreviations presented in our manuscript for better understanding. The added part could be seen in Line 527-559(marked in Red).
The paper has multiple images that need to be clarified. The images need clear figure legends. Higher-resolution images are also necessary to understand the presented data, and some figures can be removed or combined for clearer data presentation. A description of the abbreviation in a supplementary table is also needed to help readers understand what the authors are presenting.
Specifics
The authors make multiple assumptions throughout the manuscript.
Parts of Figures 1, 2, 6, 7, 8, and 10 are illegible, and Figure 1 is missing the notations.
Figure 11 needs to be clarified regarding the methodology and the findings.
Response: We have re-uploaded all figures with high resolution and clear figure legends in our revised manuscript (Figure 1-11). We also added the methodology of Figure 11 in the method and materials parts (Line 149-158 marked in Red).
We express our great appreciation again to editor and reviewers for those insightful and kind comments on our paper. Thank you for your attention and time. Looking forward to hearing from you.
Yours sincerely,
Linhui Wang
12.2022
Second Military Medical University
Shanghai, China
E-mail: wanglinhui@smmu.edu.cn; wanglinhuicz@163.com
Reviewer 3 Report
Cells-2109105- Copper Death Inducer, FDX1, as a Prognostic Biomarker Re-shaping Tumor Immunity in Clear Cell Renal Cell Carcinoma.
The authors aimed to identify novel biomarkers and targets for ccRCC management using Copper Death Inducer, FDX1. Through a retrospective study they show that FDX1 might have an impact on tumor immunity and could be a usefull tool/target for ccRCC management.
This is an interesting well-written study, thank you to share this valuable work with us and congratulations. However, based on bioinformatic analysis and without any further approve I am not really convince of its utility... yet!
I have some comments about your work:
* the quality of figure is not good enough. It is shamefull and I think you have to find a way to make it more legible.
* why do you test chemotherapy even if it is well-known that there are ineffective in ccRCC (Figure 9&10). Could it be possible to test new immunotherapy/TKI?
* in the discussion we would like you to give us your ideas/thought about future of ccRCC management... How do you see FDX1 involvement (prognostic factors on biopsy? drug testing before treatment?etc.) ?
Author Response
Dear Editor and Reviewers,
Thank you for your work and for our work titled as Copper Death Inducer, FDX1, as a Prognostic Biomarker Reshaping Tumor Immunity in Clear Cell Renal Cell Carcinoma (ID: cells-2109105). We feel sorry for the ignorance in our manuscript. According to your kind suggestions, we have made a point-to-point revision as follow:
Reviwer#3
Cells-2109105- Copper Death Inducer, FDX1, as a Prognostic Biomarker Re-shaping Tumor Immunity in Clear Cell Renal Cell Carcinoma.
The authors aimed to identify novel biomarkers and targets for ccRCC management using Copper Death Inducer, FDX1. Through a retrospective study they show that FDX1 might have an impact on tumor immunity and could be a usefull tool/target for ccRCC management.
This is an interesting well-written study, thank you to share this valuable work with us and congratulations. However, based on bioinformatic analysis and without any further approve I am not really convince of its utility... yet!
I have some comments about your work:
* the quality of figure is not good enough. It is shamefull and I think you have to find a way to make it more legible.
Response: Thank you for kind suggestions. We have re-uploaded all figures with high resolution in our revised manuscript (Figure 1-11).
* why do you test chemotherapy even if it is well-known that there are ineffective in ccRCC (Figure 9&10). Could it be possible to test new immunotherapy/TKI?
Response: Thank you for kind reminding. The reason we chose to perform drug sensitivity analysis between FDX1high and FDX1low subgroups in ccRCC is based on the poor prognosis of FDX1low subgroup, we thus chose to test the different response rate or IC50 of the chemo agents from GDSC, cellminer, CTRP and PRISM databases. As Figure9A indicated the FDX1low subgroup was more sensitive to sunitinib and crizotinib, and results from Figure 9B and 10 also reminded that FDX1 expression level was significantly correlated with the IC50 level of multi pre-clinical agents. All those findings might provide new therapeutic approaches for ccRCC patients with low FDX1 expression level. We have tested the therapeutic difference of immune related agent based on TIDE algorithm, and the results in Figure8D showed that FDX1 high expressed subgroup also led a higher response rate into immune therapy.
* in the discussion we would like you to give us your ideas/thought about future of ccRCC management... How do you see FDX1 involvement (prognostic factors on biopsy? drug testing before treatment?etc.) ?
Response: Thank you for such constructive suggestions. Since most copper death regulators (CDRs). And we added this part in discussion part in Line 503-209 (marked in red).
We express our great appreciation again to editor and reviewers for those insightful and kind comments on our paper. Thank you for your attention and time. Looking forward to hearing from you.
Yours sincerely,
Linhui Wang
12.2022
Second Military Medical University
Shanghai, China
E-mail: wanglinhui@smmu.edu.cn; wanglinhuicz@163.com
Round 2
Reviewer 2 Report
The Authors have addressed my concerns.